



**Variation in CO₂ and CH₄ Fluxes Among Land Cover Types in Heterogeneous Arctic Tundra**
**in Northeastern Siberia**
Sari Juutinen[1,2], Mika Aurela[1], Juha-Pekka Tuovinen[1], Viktor Ivakhov[3], Maiju Linkosalmi[1], Aleksi
Räsänen[4,5], Tarmo Virtanen[4], Juha Mikola[4,5] Johanna Nyman[1], Emmi Vähä[1], Marina Loskutova[6],
Alexander Makshtas[6], and Tuomas Laurila[1]
1) Finnish Meteorological Institute, Erik Palménin aukio 1, 00560 Helsinki, Finland
2) Department of Geographical and Historical Studies, University of Eastern Finland,

9       Yliopistokatu 2, FI-80100 Joensuu, Finland (P.O. Box 111, FI-80101 Joensuu, Finland)

3) Voeikov Main Geophysical Observatory, Ulitsa Karbysheva, 7, St Petersburg, 194021,

11       Russia

4) Ecosystems and Environment Research Programme, University of Helsinki, Viikinkaari 1,

13       00790 Helsinki, Finland

5) Natural Resources Institute Finland (LUKE), Latokartanonkaari 9,

15       00790 Helsinki, Finland

6) Arctic and Antarctic Research Institute, Bering str., 38, St Petersburg, 199397, Russia
Corresponding author Sari Juutinen, sari.juutinen@uef.fi


**Abstract**
Arctic tundra is facing unprecedented warming, resulting in shifts in the vegetation, thaw regimes,
and potentially in the ecosystem-atmosphere exchange of carbon (C). The estimates of regional
carbon dioxide ($CO_2$) and methane ($CH_4$) budgets, however, are highly uncertain. We measured
$CO_2$ and $CH_4$ fluxes, vegetation composition and leaf area index (LAI), thaw depth, and soil
wetness in Tiksi (71° N, 128° E), a heterogeneous site located within the prostrate dwarf-shrub
tundra zone in northeastern Siberia. Using the closed chamber method, we determined net
ecosystem exchange (NEE) of $CO_2$, dark ecosystem respiration (ER), ecosystem gross
photosynthesis (Pg), and $CH_4$ fluxes during the growing season. We applied a previously developed
high-spatial-resolution land-cover map over an m area of 35.8 $km^2$. Among the land-cover types
varying from barrens to dwarf-shrub tundra and tundra wetlands, the light-saturated NEE and Pg
scaled with the LAI of vascular plants. Thus, the graminoid-dominated tundra wetlands, with high
LAI and the deepest thaw depth, had the highest light-saturated NEE and Pg (up to -21 (uptake) and
28 mmol $m^{-2} h^{-1}$, respectively) and were disproportionately important for the summertime $CO_2$
sequestration on a landscape scale. Dry tundra, including the dwarf-shrub-dominated vegetation and
only sparsely vegetated lichen tundra, had only small $CO_2$ exchange rates. While tundra wetlands
were sources of $CH_4$, lichen tundra, including bare ground habitats, consumed atmospheric $CH_4$ at a
substantial rate. On a landscape scale, the consumption by lichen tundra and barrens could offset *ca*.
10% of the $CH_4$ emissions. We acknowledge the uncertainty involved in spatial extrapolations due
to a small number of replicates per land-cover type. This study, however, highlights the need for
distinguishing different land-cover types including the dry tundra habitats to account for their
consumption of the atmospheric $CH_4$ when estimating tundra C-exchange on a larger spatial scale.





## 1 Introduction

It is uncertain whether the Arctic tundra is a sink or a source of atmospheric carbon (C). The current estimates suggest a sink of 13–110 Tg C yr$^{-1}$, but their uncertainty range crosses the zero balance (McGuire et al. 2012, Virkkala et al. 2020). Improving these estimates is vital, because the Arctic tundra covers a vast area of 7.6 million km$^2$ (Walker 2000) that is experiencing substantial warming (IPCC 2013, Chen et al. 2021). Warming can alter C exchange, either amplifying or mitigating climate change through ecosystem–atmosphere interactions. Some local-scale studies suggest that the Arctic tundra is shifting from a small sink to a source of C (Webb et al. 2016, Euskirchen et al. 2017). It is likely that the climate change response of the ecosystem carbon dioxide ($CO_2$) sink strength and methane ($CH_4$) emissions, whether an increase or a decrease, depends on site-specific changes in thawing, wetness, and vegetation (McGuire et al. 2018). C dynamics of different tundra habitats need to be quantified across the Arctic to improve the upscaling of arctic $CO_2$ and $CH_4$ balances and to monitor how ecosystems respond to environmental changes.

The uncertainty in the arctic C balance estimates arises from the sparse and uneven observation network, which provides poor support for model-based spatial extrapolation (*cf.* McGuire et al. 2018, Virkkala et al. 2021). On a local scale, landscape heterogeneity and the related difficulty of mapping the spatial distribution of habitats and their C fluxes add to this uncertainty (McGuire et al. 2012, Treat et al. 2018, Saunois et al. 2020). In addition, year-to-year variations in seasonal features, particularly the timing of spring, summer temperatures, and snow depth have been found to cause substantial variation in the annual net $CO_2$ and $CH_4$ balances (Aurela et al. 2004, Humphreys and Lafleur 2011, Zhang et al. 2019).

Fine-scale spatial heterogeneity in soil water saturation, thaw depth, vegetation characteristics, and soil organic content is typical of the tundra landscape (*e.g.,* Virtanen and Ek 2014, Mikola et al. 2018, Lara et al. 2020). These factors control $CO_2$ and $CH_4$ exchange, and on an annual scale, tundra wetlands typically act as net $CO_2$ sinks while upland tundra areas have a close-





to-neutral C balance (*e.g.,* Marushchak et al. 2013, Virkkala et al. 2021). While tundra wetlands are
substantial sources of $CH_4$, dry tundra act as a small sink of atmospheric $CH_4$. Particularly, the
tundra barrens show high consumption rates of atmospheric $CH_4$ due to the high-affinity methane
oxidizing bacteria (Jørgensen et al. 2014, Lau et al. 2015, D'Imperio et al. 2017, Oh et al. 2020).
Thus, distinguishing dry and wet tundra with their moisture and vegetation characteristics is crucial
when mapping C exchange within the tundra biome. Treat et al. (2018) tested spatial resolution
requirements for such mapping on a landscape level and found that a 20-m pixel size captured the
spatial variation in a reasonable manner, while a coarser resolution resulted in underestimation of
both the landscape-scale $CO_2$ uptake and $CH_4$ emissions. In addition, understanding the spatial
heterogeneity of ecosystem C exchange substantially enhances analyses of micrometeorological
measurements that, while in principle representing spatially integrated fluxes, may provide biased
balances in a highly heterogeneous environment (*e.g.,* Tuovinen et al. 2019). Thus, plot-scale data,
allowing studies of the local relationship between gas exchange and soil and vegetation properties,
are necessary for improving and validating upscaling methods.

The aim of this study was to assess the spatial patterns and magnitudes of $CO_2$ and

$CH_4$ fluxes within heterogenous prostrate dwarf-shrub tundra in Tiksi, located in northeastern
Russia. Growing season fluxes of $CO_2$ (ecosystem net exchange, photosynthesis, and respiration)
and $CH_4$ were determined using the chamber method to answer the questions: (i) what is the
magnitude of these fluxes in different land-cover types? And (ii) how do they depend on vegetation
characteristics and soil wetness? In addition, to test the spatial representativeness of the chamber
data, we extrapolated the habitat-level measurements in space to compare them with the ecosystem-
level data measured with the micrometeorological eddy covariance (EC) technique.







## 2 Materials and Methods

### 2.1 Study site

The study site is located near the Tiksi Observatory (see Uttal et al. 2016) in Yakutia, northeastern Russia (71.5943 N, 128.8878 E), 500 m inland off the Laptev Sea coast and, on average, 7 m above sea level (Fig. 1). The area belongs to the middle-arctic prostrate dwarf-shrub tundra subzone (Walker, 2000) and has continuous permafrost. In the end of the growing season, the maximum thaw depth is 40 cm (Mikola et al. 2018). Climate in Tiksi is defined by cold winters and cool summers. The long-term mean annual temperature and mean annual precipitation were -12.7 °C and 232 mm, respectively, during the normal period 1981–2010. Growing season lasts about 3 months, and the soils typically freeze in the end of September and the permanent snow falls in October and thaws in June (AARI 2018).

Soil organic content varies from negligible in lichen covered and bare graveled areas to *ca.* 40% in tundra wetlands (Mikola et al. 2018). Bedrock and soils are alkaline, resulting in high plant species richness. Vegetation consists of mosses, lichens, grasses, sedges, prostrate dwarf-shrubs such as willows (*Salix* spp.), dwarf birch (*Betula nana*), and *Diapensia lapponica,* and forb species (Fig. 1, Table 1). The average heights of dwarf-shrub species are 4–6 cm and the leaf area index (LAI) of vascular plants reaches up to 1 $m^2\ m^{-2}$ in the wetland and meadow habitats with graminoid vegetation (Juutinen et al. 2017). The land cover at the site has been classified *a priori* and mapped based on a combination of field inventories and high-spatial resolution satellite images (Mikola et al. 2018). The *a priori* land-cover types (LCT) consist of wet fen, dry fen, graminoid tundra, bog, meadow at the stream bank, dwarf-shrub tundra, and lichen tundra (includes bare ground with vegetation patches) (Table 1). A section of the wet and dry fen within the EC footprint area is disturbed by vehicle tracks that create open water surfaces, and there is also an area of eroded bare-peat surface on a dry fen.





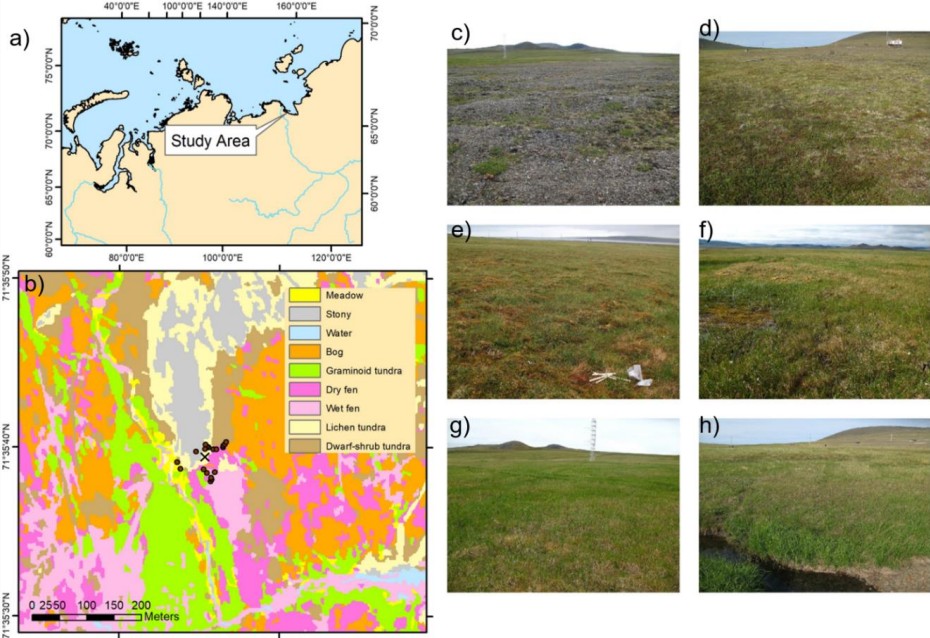

**Fig. 1. a)** Location of the study area in Tiksi, Yakutia, Russia, **b)** Land-cover map and the chamber flux measurement points (dots) and the EC mast (×) on the map, and photos of the LC types: **c)** lichen tundra with barrens, lichens, and patches of vegetation, **d)** dwarf-shrub tundra, **e)** bog, **f)** wet and dry fen, **g)** graminoid tundra, and **h)** meadow by the stream.














**Table 1.** Soil and vegetation characteristics of the land cover types (LCT) and their proportions in
the EC impact area (90% of the cumulative footprint).

| LCT | Soil properties and plant taxa | Proportion (%)[2] |
|---|---|---|
| Lichen tundra[1] | Mixture of vegetated patches, stones, and bare ground. Lichens, *Dryas octopetala, Vaccinium vitis-ideae, Salix polaris, Diapensia lapponica, Oxytropis* spp, *Astragalus* spp., *Pedicularis* spp., *Artemisia* spp., *Minuartia* sp., | 8 (barren), 11 (sparse vegetation) |
| Dwarf-shrub tundra | Shallow organic layer on mineral soil ground Feather mosses, lichens, *Salix polaris, Vaccinium vitis-ideae, Vaccinium uliginosum, Dryas octopetala, Cassiope tetragona, Betula nana, Polygonum viviparum, Pedicularis* spp., *Carex* spp. | 18 |
| Meadow | Shallow organic layer on mineral soil ground *Calamagrostis* sp., *Festuca* sp, *Salix* spp. *Polygonum viviparum, Bistorta major, Polemonium* sp., *Valeriana* sp. | 1.4 |
| Graminoid tundra | Shallow peat layer on mineral soil ground Feather mosses, *Sphagnum* spp., *Carex* spp., *Eriophorum* spp., *Calamagrostis* spp., *Salix* spp., *B. nana, Saxifraga* spp., *Ranunculus* spp., *Bistorta major, Stellaria* sp., *Valeriana* sp., *Polemonium* sp., *Comarum palustre* . | 13 |
| Bog | Dry hummock habitat at the tundra peatland *Sphagnum* spp., feather mosses, *Salix* spp., *Vaccinium uliginosum, Vaccinium vitis-idaea, Betula nana, Rhodendron tomentosum, Cassiope tetragona, Carex* spp., *Polygonum viviparum., Stellaria* sp. | 23 |
| Dry fen | Intermediate wet tundra peatland habitat *Sphagnum* spp., *Carex* spp., *Salix* spp *Saxifraga* spp., *Comarum palustre, Epilobium* spp., *Ranunculus* spp., *Pedicularis* spp., *Stellaria* sp. | 10 |
| Wet fen | Wet tundra peatland habitat with open pools *Brown mosses, Carex* spp., *Eriophorum* spp., *Ranunculus* sp., *Caltha palustris, Pedicularis* sp., *Saxifraga* sp. | 15 |

[1] Combined land-cover types bare and lichen tundra in Juutinen et al. (2017), Mikola et al. (2018),
Tuovinen et al. (2019), [2] Proportion within the 90% coverage of the mean EC footprint area during
the growing season of 2014 (Tuovinen et al. 2019).








*2.2 CO$_2$ and CH$_4$ flux measurements*
Fluxes of CO$_2$ and CH$_4$ were measured using static chambers set on 12 pre-installed collars of 50
cm × 50 cm. The measurement points (collars) were set to cover the heterogeneity in land cover,
and in each study year, there were 1–4 measurement points per each LCT (Table 2). Most of the
data were collected during a study campaign in July 15 – August 16, 2014. The growing season had
started earlier due to a warm period and daily mean air temperature stayed over 5 °C since July 5
(Fig. 2) (Tuovinen et al. 2019). Net ecosystem exchange of CO$_2$ (NEE) and ecosystem respiration
of CO$_2$ in dark (ER) were measured using transparent and opaque chambers (transparent chamber
covered with a hood), respectively, allowing the estimation of ecosystem gross photosynthesis (Pg)
as difference of NEE and ER. Fluxes of CH$_4$ were determined from closures of both transparent and
opaque chambers, but because there was no difference between them when performed
consecutively, the data from transparent chamber measurements were used for flux calculations. In
addition, CH$_4$ fluxes were measured during shorter campaigns in 2012, 2013, 2016, and 2019
(Table 2). These data also included vehicle track disturbance plots and an eroded bare-peat surface,
which were measured in 2019.
**Table 2.** Measurement periods, measured fluxes (CH$_4$, ER, NEE), and number of measurement
points and observations (points, observations) in each land cover type (LCT) across the study years.

| LCT | 2012 Jul 18–21 | 2013 Jul 5–Sep 3 | 2014 Jul 15–Aug 16 | 2016 May 30, Aug 4–5, Sep 13–14 | 2019 Aug 28–Sep 1 |
|---|---|---|---|---|---|
| | CH$_4$ | CH$_4$ | ER, NEE, CH$_4$ | CH$_4$ | CH$_4$ |
| Wet fen | 4, 4 | 6, 22 | 3, 107 | 3, 27 | 5, 72 |
| Vehicle track | | | | | 2, 30 |
| Dry fen | 2, 2 | 4, 11 | 3, 107 | 3, 14 | 2, 26 |
| Bare peat | | | | | 1, 15 |
| Bog | 2, 2 | 3, 7 | 1, 36 | | 1, 13 |
| Meadow | 1, 1 | 2, 6 | 2, 62 | | |
| Dwarf-shrub tundra | 1, 1 | | 1, 36 | 1,1 | |
| Lichen tundra | | 1, 3 | 2, 67 | 2, 18 | 2, 29 |
| Snow and ice[1] | | | | 2, 2 | |

[1]Measured only on May 30, 2016.



In 2012 and 2013, $CH_4$ concentrations inside the chamber were analyzed from
samples stored in glass vials using a gas chromatograph equipped with a flame ionization detector
in the laboratory of the Voeikov Main Geophysical Observatory. Four samples per each 20-min
chamber closure were collected. Since July 2014, $CH_4$ and $CO_2$ concentrations inside the chambers
were recorded every second during closures of about 5-min using a gas analyzer (Los Gatos
Research, DLT-100). Gas fluxes between the ecosystem and the atmosphere were calculated from
the phase of linear concentration change in the chamber head space over time accounting for
temperature, volume, and atmospheric pressure. Concentration change during each chamber closure
was evaluated visually for determining the closure start time and to remove cases showing
nonlinearity due to leaks, ebullition, or saturation. There were a few ebullition cases at the vehicle
track measurement points that had only sparse or no vegetation cover.
The fluxes of $CO_2$ and $CH_4$ were also measured by the micrometeorological EC
method, which provides continuous data of the atmosphere-biosphere fluxes averaged on an
ecosystem scale. The EC system consisted of a three-dimensional sonic anemometer (USA-1,
METEK Gmbh, Elmshorn, Germany), a closed-path $CH_4$ analyzer (RMT-200, LGR, Inc., CA,
USA), and a closed-path $CO_2/H_2O$ analyzer (LI-COR LI-7000, Inc., Lincoln, NE, USA). The fluxes
were calculated as 30-min averages and processed using standard methods (Aubinet et al. 2012).
The EC measurement system and the post-processing procedures have been presented in more
detail by Tuovinen et al. (2019).
Supporting meteorological measurements including air temperature (Tair) (Vaisala,
HMP), soil temperature (Tsoil) (IKES, Nokeval), photosynthetic photon flux density (PPFD) (Kipp
& Zonen, PQS1), and water table level relative to the ground surface (WT) (8438.66.2646, Trafag)
were collected by a Vaisala QML datalogger as 30-min averages. We also present meteorological
data for the period 2011–2019 to relate the conditions during the measurement campaign in Jul 15-





Aug 16, 2014, and the $CH_4$ flux campaigns in 2012, 2013, 2014, 2016, and 2019, to the nine-year
overall.

*2.3 Vegetation and Topographic Wetness Index*
On a site level, vegetation and soil characteristics were inventoried in plots assigned into a
systematic grid outside the area covered by the gas flux measurement points in 2014 (see Juutinen
et al. 2017; Mikola et al. 2018). The projection cover (%) of plant species and species groups, and
the mean canopy height of each species group were recorded. Seven species groups were included
in the inventory: *Sphagnum* mosses, feather mosses, brown mosses, dwarf shrubs, *Betula nana*,
*Salix* species, forbs, and graminoids. A subset of the plots was harvested, and vascular plant leaves
were scanned to determine the one-sided LAI to estimate empirical relationships between LAI and
%-cover and canopy height to estimate LAI in the collars (see Juutinen et al. 2017). In the collars,
cover (%) and height (cm) of each species group were recorded weekly during the gas flux
measurement campaign July 15–August 16, 2014. Because there were no observational vegetation
data for the other years than 2014, the green chromatic coordinate (GCC) was used as a proxy for
the amount of green above-ground vascular plants (*e.g.* Richardson 2019). GCC was calculated
from the digital numbers of red (R), green (G), and blue (B) color channels as the ratio of green in
the images (GCC= G/(R+G+B)) from digital RGB photos of the vegetation inside the collars. The
photos were taken at the time of measurements. We determined an empirical relationship between
LAI and GCC by using a data set of harvested plots with digital RGB photographs and measured
LAI data (n=91). Data distributions varied among the LCTs due to the intrinsic differences and
amount of vegetation. For the LAI estimation, we used a linear relationship ($R^2 = 0.46$, p<0.001)
between LAI and GCC determined using the entire data set (see appendix Fig. 1 for the data and
equation).





To quantify potential soil wetness at each measurement point, we calculated the mean

topographic wetness index (TWI) value based on a 2 m spatial resolution digital elevation model
(Mikola et al. 2018). To characterize differences between growing seasons as manifested by
vegetation greenness, MODIS Normalized Difference Vegetation Index (NDVI) with 16-day
temporal and 500 m spatial resolution was calculated for a circular area with 300 m radius from the
flux tower using Google Earth Engine (Gorelick et al. 2017). NDVI was derived for 2011–2019 to
place the measurement years in the context of year-to-year variation in weather.

*2.4 Data analyses*
When examining the role of the habitat types in $CO_2$ and $CH_4$ exchange, we applied the land cover
classification presented in Mikola et al. (2018). The data collected in July 15 – August 16, 2014
were used for examining gas exchange in relation to the variation in LAI, GCC, WT, and TWI
among the collars. Utilizing the ER and NEE fluxes measured with opaque and transparent
chambers, respectively, we assessed the light response of Pg and NEE with a hyperbolic function

$NEE = ER - Pg_{max} \times PPFD/(\beta+PPFD),$                     eq. 1.
where $Pg_{max}$ is the asymptotic maximum of photosynthesis, and $\beta$ is the half-saturation PPFD. To
ensure comparability between different measurement days in relatively low light conditions, we
determined the light-normalized $Pg_{800}$, *i.e.,* $Pg$ at PPFD = 800 $\mu$mol m$^{-2}$ s$^{-1}$. The corresponding
NEE, *i.e.,* $NEE_{800}$, is then obtained as a sum of $Pg_{800}$ and ER. Fluxes of $CH_4$ are expressed as collar
means. We used a sign convention where a positive value means net release to the atmosphere and a
negative value denotes net uptake by the ecosystem.

To find the main factors and gradients in the plant community, gas flux, and

environmental variables data measured in the flux collars in 2014, we performed a detrended
correspondence analysis (DCA) of the species group data with post-hoc fit of environmental



variables, including gas fluxes, WT, LAI, GCC, elevation, and thaw depth as supplementary
variables. The DCA was performed on logarithmically transformed, centered species data (species
as species groups) using Canoco 5 (Ter Braak and Šmilauer 2012). Regression analyses were used
to test the relationships between gas flux estimates and vascular LAI, GCC, WT, and TWI. All $CH_4$
flux data from the years 2012–14, 2016, and 2019 were used to quantify the mean growing season
$CH_4$ flux for each LCT and examine the relationship between $CH_4$ and GCC and TWI.

We compared the LCT-specific flux estimates based on the chamber measurements

with the estimates based on EC measurements over the same period. Partitioning of the EC-based
$CO_2$ fluxes to Pg and ER and estimates of $Pg_{800}$ and $NEE_{800}$ were calculated similarly to that of
chamber data using Eq. (1). The EC flux data were classified into five wind sectors (30–125°, 125–
185°, 185–239°, 239–310°, 310–360° based on the mean EC flux footprint, modeled for the
growing of 2014 by Tuovinen et al. (2019). The sectors distinguished areas dominated by different
LCTs, especially tundra heaths and wetlands, and, similarly, sectors with large and small vascular
LAI. For each sector, the footprint-weighted areal proportions of LCTs and mean vascular LAI
were derived from the high spatial resolution land-cover and LAI maps (Mikola et al. 2018). For
this comparison, sector averages of $Pg_{800}$, ER, $NEE_{800}$, and $CH_4$ flux were calculated from the
chamber data by weighting the LCT-specific flux estimates with the above-mentioned LCT
proportions in each sector. Because there were no measurement points within graminoid tundra, we
applied wet fen (for $CO_2$) and dry fen (for $CH_4$) flux estimates for the graminoid tundra based on
the observed similarities in LAI and soil wetness, respectively. Overall, graminoid tundra can be
considered part of the fen continuum in terms of soil characteristics (high organic content) and $CH_4$
exchange (Mikola et al. 2018, Tuovinen et al. 2019).

Finally, to synthesize the $CO_2$ and $CH_4$ exchange variability across the tundra, we

upscaled the LCT-specific average $NEE_{800}$, $Pg_{800}$, ER, and $CH_4$ flux to the 35.8 $km^2$ area
surrounding our study site, for which a LCT map was produced by Mikola et al. (2018).



## 3 Results

*3.1 Meteorology*

In 2014, when we collected most of the flux data, temperature sum accumulation (with a 0 °C $T_{air}$ threshold) was near-average during the thaw period (the period when soil surface temperature was continuously above 0°C), but the spring and mid-growing season were warmer than on average (Fig. 2a). The average air temperature was 15°C during the gas flux measurements. Accordingly, the MODIS NDVI showed an early start of greening (Fig. 2b-d), and vegetation development had already started at the beginning of the measurement period. In 2010–2019, which included the other $CH_4$ measurement years, the thaw period lasted for 74–124 days, creating a temperature sum range of 642–1003 °C days (Fig. 2a). Surface soils thawed between May 28 and July 9 and froze again between September 21 and October 1. Among the observation years, the years 2012 and 2019 had notably longer and warmer thaw periods than the other years. The driest habitat, lichen tundra, thawed 10–15 days earlier than the other habitats, and had ca. 3 °C higher soil temperature than the wet fen at the depth of 5 cm (Fig. 2b–c). Water table depth, measured at a wet fen location, showed only subtle interannual variation (Fig. 2e). In 2014, the active layer depth, measured over the measurement period close to the collars, was deepest in the end of August, reaching 30–40 cm in the wetland and meadow habitats, and remained < 30 cm in the dry dwarf-shrub tundra (Fig. 2f). Lichen tundra had rocks underneath the loose surface layer, which made it impossible to measure the actual thaw depth.



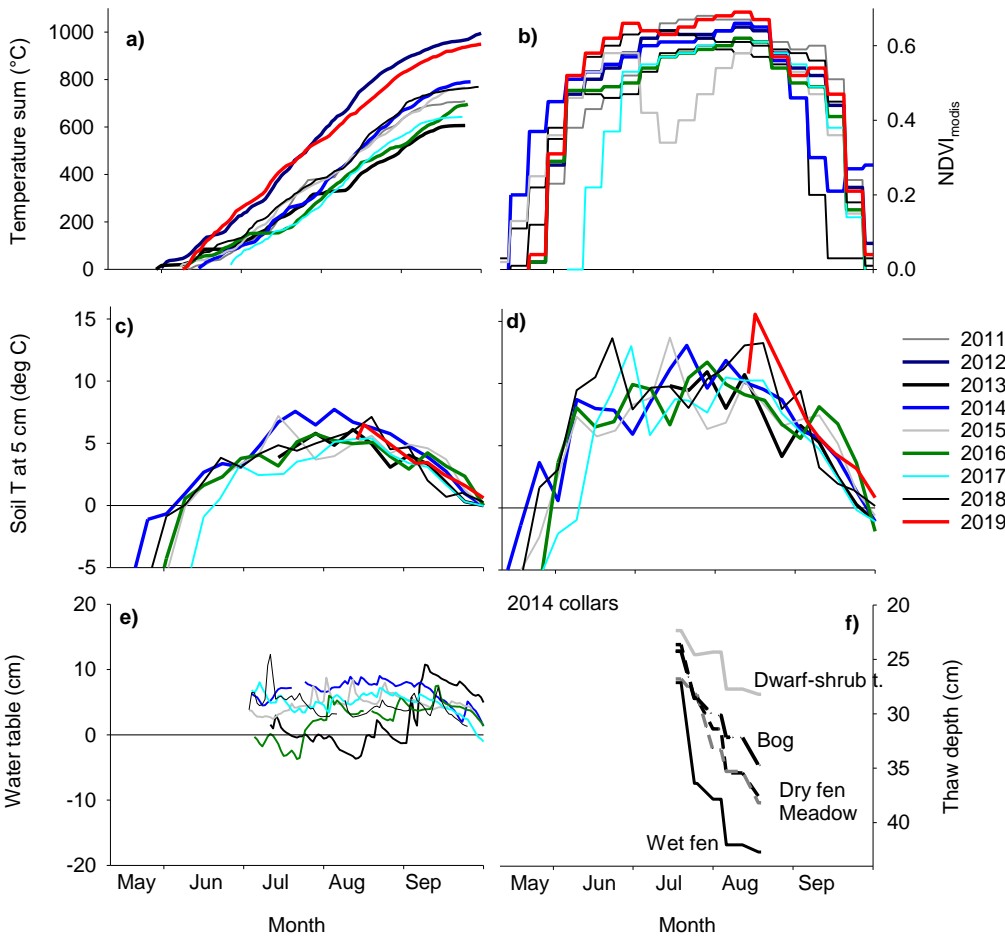

**Fig. 2.** Meteorology in May to September in years 2011–2019. (**a**) Air temperature accumulation with threshold values soil surface > 0 °C and air T > 0 °C, **b**) seasonal dynamics of NDVI in the study area, 16 d aggregated MODIS data, **c**) weekly means of soil temperature at depth of 5 cm in wet fen and **d**) in dry tundra, **e**) water table relative to the ground surface in wet fen, and **f**) LCT means of thaw depth in the measurement collars in 2014.

### 3.2 *Multivariate analysis*

DCA axes 1 and 2, which explained 49% and 14% of the variation in the grouped species data (Fig.

3). Lichen tundra and wet fen plots differed most from each other along the axis 1. Accordingly, the

supplementary variables WT, vascular plant LAI, thaw depth, TWI, GCC, $Pg_{800}$, $NEE_{800}$, and $CH_4$



fluxes correlated positively with the axis 1 with post-hoc correlations (r) of 0.6–0.9, as derived from
the DCA weighted correlation matrix. Elevation had a positive correlation with the axis 2 (r = 0.8),
along which there were gradients in moss abundance and soil organic content.


**Fig. 3.** DCA ordination diagram based on species (species groups) data from the measurement
collars in 2014. In the plot, the scores of species groups (cross), sample plots (open symbols), and
post-hoc fits of supplementary variables (arrows, blue type) mean $CH_4$, $Pg_{800}$, ER, $NEE_{800}$, thaw
depth (Thaw), water table relative to the ground surface (WT), green chromatic coordinate (GCC),
vascular plant LAI, and elevation above sea level (Elevation). Land-cover types of the sample plots
are indicated (grey type) and plots assigned to same LCTs are circled. Eigenvalues for axis 1 and 2
are 0.597 and 0.171, respectively, and axis 1 and 2 explain cumulatively 63% of the variation in the
species group data.





3.3  *Exchange of CO₂ and CH₄*
Among different LCTs, the light-normalized photosynthesis ($Pg_{800}$) varied from about 5 mmol m$^{-2}$
h$^{-1}$ in the lichen tundra to about 22 and 27 mmol m$^{-2}$ h$^{-1}$ in the wet fen and meadow, respectively.





$Pg_{800}$ was strongly and positively related to the vascular plant LAI and the greenness index GCC
(Fig. 4). There was also a positive correlation between $Pg_{800}$ and WT and TWI, possibly because the
highest LAI occurred at the wet fen and meadow plots. However, the TWI values for the two
meadow plots located on an elevated bank of the stream were disproportionately high in relation to
the WT at the plots, probably because of insufficient locational accuracy or an artefact in the digital
elevation model. Ecosystem respiration was higher in the two meadow plots, on average 18 mmol
$m^{-2} h^{-1}$ than in other plots. The relationship between ER and LAI was weaker than that of $Pg_{800}$ and
LAI (Fig. 4). The net exchange, $NEE_{800}$, varied from about zero in the lichen tundra plots to a net
$CO_2$ uptake of 16 mmol $m^{-2} h^{-1}$ in the meadow and wet fen plots. $NEE_{800}$ was more tightly linked to
$Pg_{800}$ than ER and was correlated with LAI, GCC, WT, and TWI (Fig. 4).

There was substantial consumption of the atmospheric $CH_4$ in the lichen tundra plots

(mean -0.02 mmol $m^{-2} h^{-1}$, Fig. 5). Minor consumption occurred in the meadow, dwarf-shrub tundra,
and bog plots (mean <-0.002 mmol $m^{-2} h^{-1}$), and efflux to the atmosphere was observed in the dry
fen and wet fen plots (means 0.05 and 0.16 mmol $m^{-2} h^{-1}$, respectively, Fig. 5). The eroded bare-peat
plot within the dry fen habitat and the vehicle-track plots in wet fen had large emissions (up to 0.2
mmol $m^{-2} h^{-1}$), which were of the same magnitude as in the undisturbed dry and wet fen habitats.
Variation among the plot means (Fig. 4, year 2014 data) was positively correlated with WT. Large
$CH_4$ emissions occurred when TWI was > 4, except the two meadow plots, which showed net
consumption of $CH_4$ but had an unrealistically high TWI (see above and Figs. 4 and 6). Variation in
$CH_4$ fluxes was not related to variation in LAI or GCC.




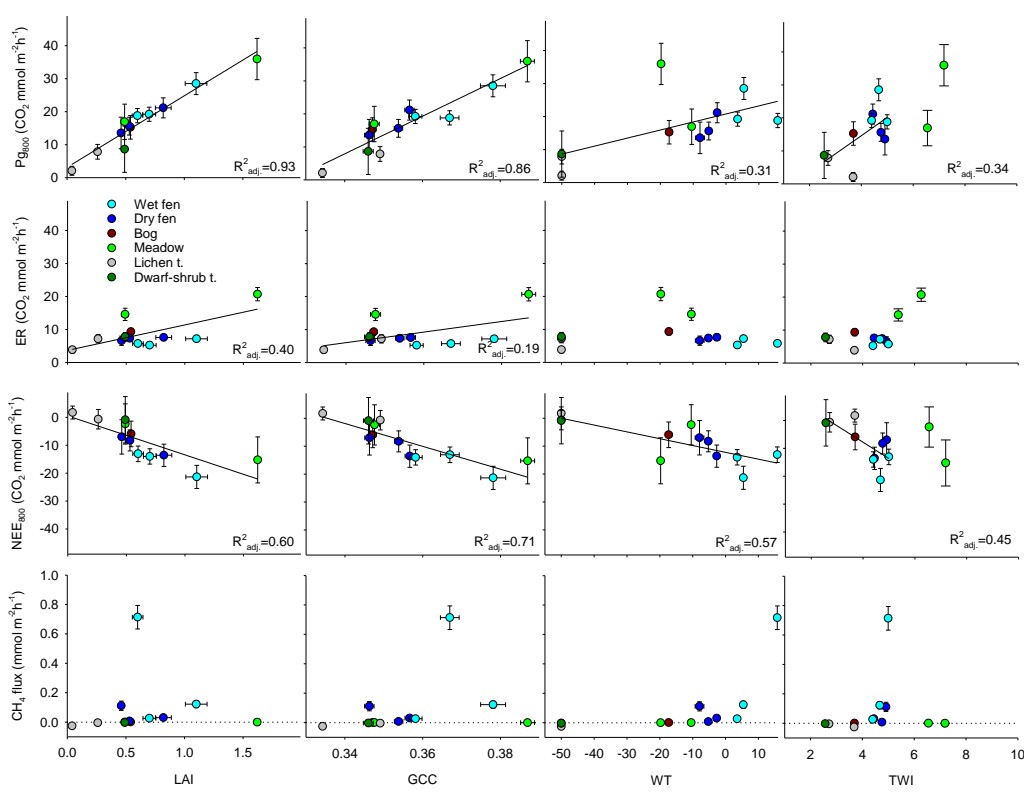

**Fig. 4.** Variation in estimates of $Pg_{800}$, ER, $NEE_{800}$ (Eq. 1) and collar means of $CH_4$ fluxes in relation to variation in collar means of LAI, GCC, WT and TWI on July 6–August 16, 2014. Error bars denote the standard error of estimate. Fitted regression lines and adjusted coefficients of determination ($R^2_{adj.}$) are included for significant linear relationships. The two meadow plots were not included in the TWI regressions.





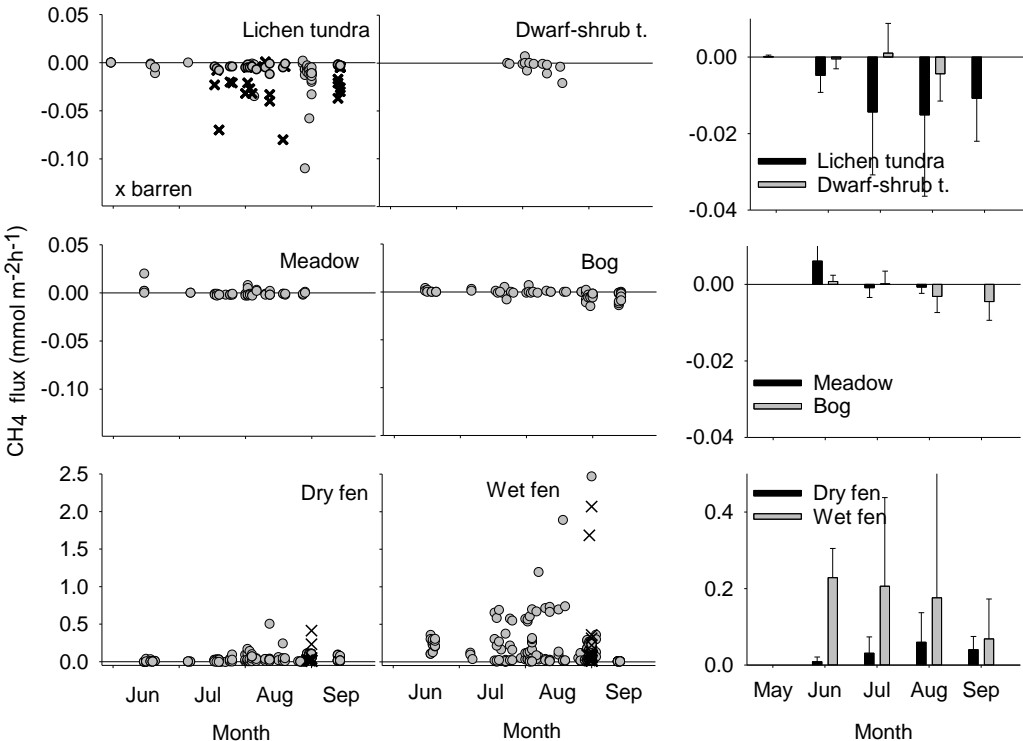

**Fig. 5.** Instantaneous (left panels) and monthly mean (right panels, with ±SD error bars) CH₄ fluxes
in each LTC. The data are a composite of all study years. Barren surfaces are indicated among the
lichen tundra data. The eroded bare-peat and vehicle-track plots are plotted as part of the dry fen
and wet fen data (×), respectively, but these data are not included in the monthly means. Note that
the panel groups have different y-axis scales.



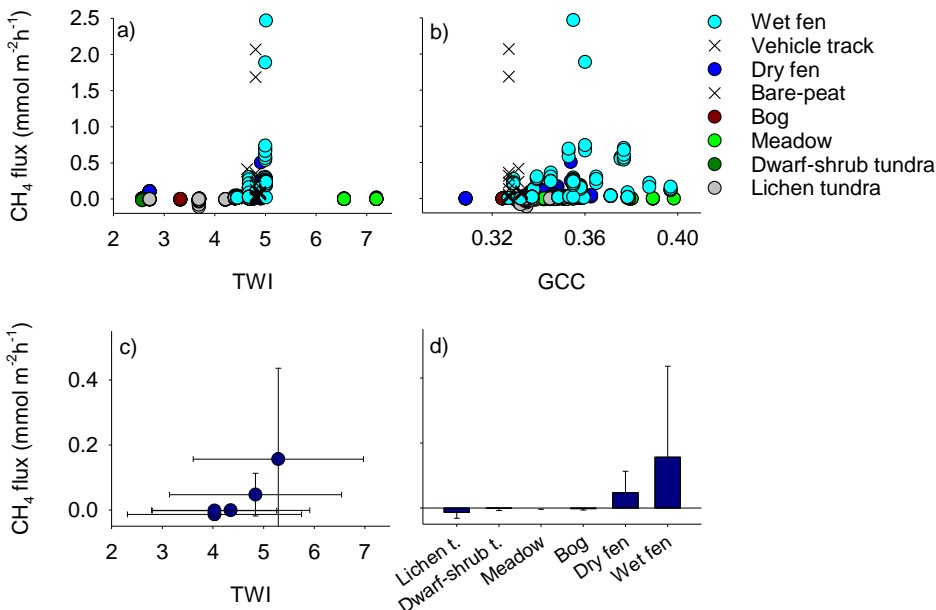

**Fig. 6.** Instantaneous $CH_4$ fluxes in the LCTs in relation to **a)** plot specific TWI and **b)** GCC and **c)** LCT mean (±SD) CH4 fluxes in relation to LCT mean (±SD) TWI (excluding the meadow plots with erroneous TWI) and **d)** LCT mean $CH_4$ fluxes (±SD). Data from years 2014, 2016, and 2019.

To compare the chamber-based flux data with those derived from the EC measurements, the EC data were classified based on wind direction, which reflects the varying domination of different LCTs within the EC source area. In both the southern and south-western wind sectors (125–185° and 185–239°), wet fen and graminoid tundra together contributed *ca.* 40% of the footprint-weighted LCT areas (Fig. 7a). In these directions, vegetation mainly consisted of graminoids, as dry fen, wet fen, graminoid tundra, and meadow contributed 80% in total. The northern sector (310–360°) was characterized by the abundance of lichen tundra and bare ground that accounted for 68% of the footprint-weighted LCT areas, while all the other LCTs covered less than 18% in total. The other wind direction sectors had more even LCT distributions. The differences between the sectors were similar in the EC-based and spatially weighted chamber-based averages of $CO_2$ exchange (Fig. 7). Both $Pg_{800}$ and $NEE_{800}$ were largest in the southern and south–



western sectors and clearly smallest in the barren–lichen tundra-dominated sector in the north. The
chamber-based estimates of $CO_2$ exchange were, however, lower: $Pg_{800}$ was 57%, ER was 93%, and
$NEE_{800}$ was 44% of the EC-based estimate.

The southern and south-western wind sectors with abundant dry and wet fens and

graminoid tundra had clearly the largest $CH_4$ fluxes (Fig. 7). The estimate based on chamber
measurements was 30% and 50% larger than the EC-based estimate for the east sector (dominated
by dry fen and bog) and south sector (dominated by dry and wet fen), respectively. In contrast, the
chamber-based estimate was 56–67% of the EC-based estimate for the other sectors, dominated by
graminoid tundra and lichen tundra.

Within the extended study area of 35.8 km$^2$, the LCT-weighted mean $NEE_{800}$,

corresponding to the LCT-specific chamber-based fluxes that were upscaled with the footprint-
weighted LCT areas, was -6 mmol m$^{-2}$ h$^{-1}$ (uptake relative to the atmosphere). The corresponding
mean $Pg_{800}$ was 12 mmol m$^{-2}$ h$^{-1}$, and $CH_4$ flux 0.05 mmol m$^{-2}$ h$^{-1}$ (Table 3). Relative to their spatial
cover (28% in total), wet and dry fens were disproportionally important for the landscape-level net
exchange of $CO_2$, photosynthesis, and $CH_4$, contributing 74%, 47%, and 99% of the net landscape
totals (Table 3). Consumption of $CH_4$ by lichen tundra (including barrens), dwarf-shrub tundra, and
meadow tundra soils was 10% of the $CH_4$ emission. Particularly, the barrens contributed to the
consumption of $CH_4$ due to their large area and high consumption rate. Note, however, that the EC-
based estimates for the wind direction sectors suggested about two times as high $NEE_{800}$ and *ca.*
30% smaller $CH_4$ emissions for the wet fens, and 30% larger consumption for the barrens and
lichen tundra.

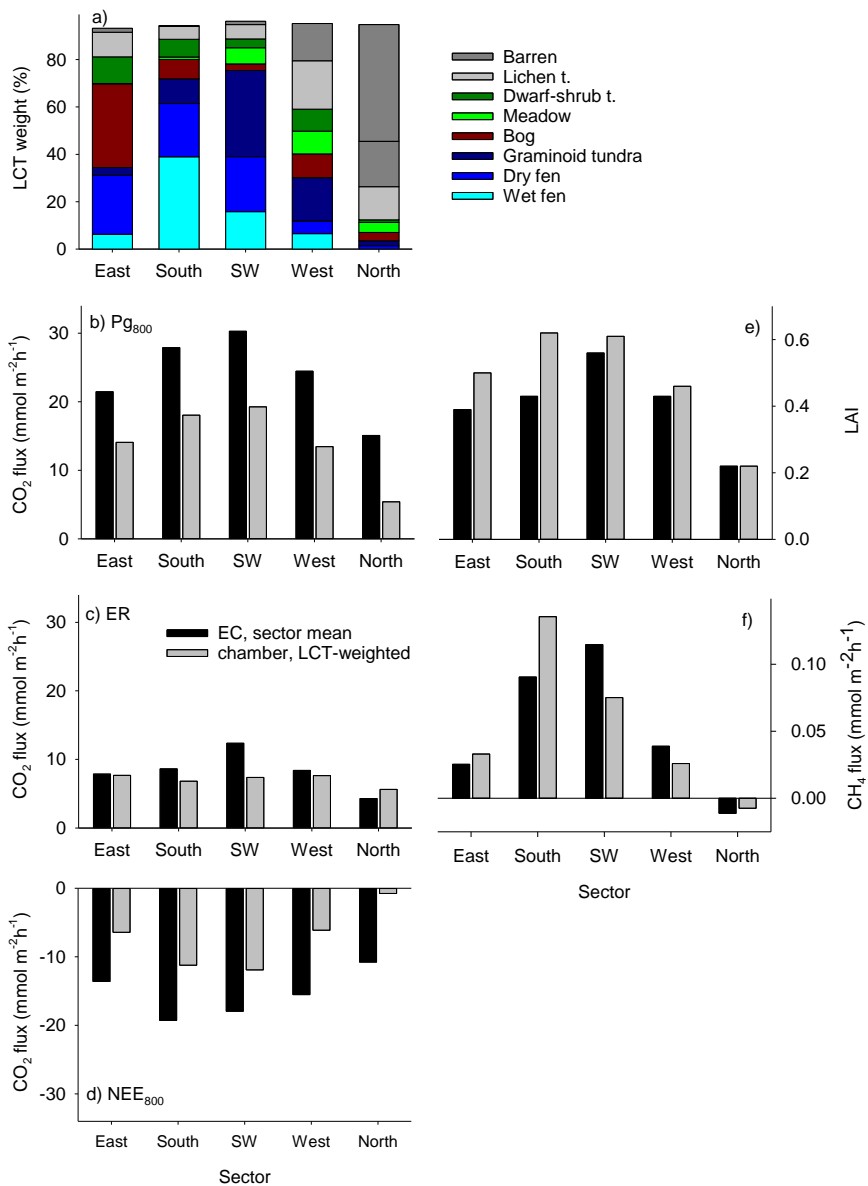

**Fig. 7. a)** Footprint-weighted mean contribution of each LCT to the EC measurements divided into wind direction sectors, and comparison of EC and chamber-based sector means of **b-d)** $CO_2$ exchange ($Pg_{800}$, ER, and $NEE_{800}$) **e)** vascular plant LAI, and **f)** $CH_4$ fluxes. The chamber-based data are weighted by the LCT proportions shown in panel a. Map of LAI (Tuovinen et al., 2019) and the LAI measured in the collars were used to estimate the EC- and chamber-related sector means, respectively.














**Table 3.** Land-cover type distribution in the mapped 35.8 km$^2$ area, means and standard errors (se)
calculated from the collar-specific estimates of $Pg_{800}$, ER, $NEE_{800}$ and $CH_4$ based on the 2014 data,
and proportional (%) landscape budgets.

| LCT | Area (%) | $Pg_{800}$ (mmol m$^{-2}$h$^{-1}$) | | ER (mmol m$^{-2}$h$^{-1}$) | | $NEE_{800}$ (mmol m$^{-2}$h$^{-1}$) | | $CH_4$ flux (mmol m$^{-2}$h$^{-1}$) | | $NEE_{800}$ (%) | $Pg_{800}$ (%) | $CH_4$ flux (%) |
|---|---|---|---|---|---|---|---|---|---|---|---|---|
| | | mean | sd | mean | sd | mean | sd | mean | sd | | | |
| Mean[1] | | 11.21 | | 6.60 | | -4.61 | | 0.053 | | | | |
| Wet fen | 16.4 | 21.93 | -3.91 | 6.44 | 0.98 | -15.49 | 3.39 | 0.223 | 0.286 | 55.1 | 32.1 | 88.66 |
| Dry fen | 11.6 | 14.60 | 1.02 | 6.99 | 0.39 | -7.61 | 0.64 | 0.059 | 0.052 | 19.1 | 15.1 | 10.80 |
| Gram. t. | 3.4 | 21.93 | 3.91 | 6.44 | 0.98 | -15.49 | 3.39 | 0.059 | 0.052 | 11.4 | 6.7 | 10.77 |
| Bog | 9.1 | 15.27 | | 9.34 | | -5.93 | | 0.000 | | 11.7 | 12.4 | 0.03 |
| Meadow | 0.4 | 26.45 | 9.51 | 17.66 | 3.06 | -8.79 | 6.45 | -0.001 | <0.001 | 0.8 | 0.9 | -0.01 |
| Dwarf-s. t. | 27.4 | 8.64 | | 7.80 | | -0.85 | | -0.003 | | 5.0 | 21.1 | -1.65 |
| Lichen t. | 11.1 | 4.98 | 2.87 | 5.53 | 1.68 | 0.55 | 1.20 | -0.005 | | -1.3 | 4.9 | -1.05 |
| Barren | 15.3 | 4.98 | 2.87 | 5.53 | 1.68 | 0.55 | 1.20 | -0.026 | | -1.8 | 6.8 | -7.56 |
| Water | 5.3 | NA | | NA | | NA | | NA | | | | |

[1] Spatial LCT area-weighted mean, [2]Graminoid tundra fluxes estimated using values for wet fen
($CO_2$) and dry fen ($CH_4$)


## 4   Discussion

The studied tundra site in Tiksi in northeastern Siberia has heterogeneous land cover, which is
reflected as equally heterogeneous $CO_2$ and $CH_4$ exchange. On the one hand, we found that the
tundra wetlands have a disproportional role: dry and wet fens and meadow had the highest $CO_2$
uptake capacity and particularly the wet fen showed high $CH_4$ emissions. On the other hand, our
results highlight the high consumption of atmospheric $CH_4$ by lichen tundra (barrens and small
vegetated patches). This $CH_4$ consumption is high compared to other non-wetland tundra habitats
and, on the landscape scale, could offset 9 of the $CH_4$ emissions. These data augment the
knowledge on the functional diversity, namely distribution of different land-cover types, and their
emission factors across the vast arctic tundra and will lend support to bottom-up and top-down
extrapolations across the Arctic.

Within this tundra landscape, the graminoid-dominated wetlands with organic-rich soils

constitute an important part of the ecosystem-atmosphere exchange of $CO_2$ and $CH_4$. Within an area





of 35.8 km$^2$ mapped around our study site (Mikola et al. 2018), wet and dry fens and the fen-like
graminoid tundra covered 31% of the area but contributed as much as 73% to the potential light-
saturated $CO_2$ sink during the peak growing season. These wetlands are also the sites having high
soil organic matter content and C pools (Mikola et al. 2018) and $CH_4$ emissions to the atmosphere
(see also Tuovinen et al. 2019).

The spatial extrapolation of fluxes is clearly sensitive to a small number of chamber

measurement points as there is large within-LCT variation e.g., in the wet fen and meadow data. For
this reason, it is neither possible to conclude which LCTs differ significantly from each other in the
$CO_2$ or $CH_4$ fluxes. Our conclusions made from the chamber data are, however, corroborated by the
temporally matching section of EC data, categorized by wind direction to reflect the main LCT
patterns around the EC mast, which show similarity to the chamber data. Instead of categorical LCT
classification, maps of those variables, LAI, and WT, for instance, that drive $CO_2$ and $CH_4$
exchange would be preferable for spatial modeling of these ecosystem functions (Räsänen et al.
2021). Mikola et al. (2018) found, however, that distinguishing, for instance, soil organic content
based on remote sensing and using the same LCT classification was a challenge in the same site.

The spatial pattern of the growing season light-saturated photosynthesis and net $CO_2$

exchange was strongly related to the corresponding pattern of the LAI of vascular plants (Fig. 3, 4).
Hence, the abundance of graminoid (Cyperaceae and Poaceae) vegetation predicted a large $NEE_{800}$,
which varied from near zero in lichen tundra up to 25 mmol m$^{-2}$ h$^{-1}$ in wet fen. Ecosystem
respiration had a smaller role than Pg in determining NEE, but we note that our data cover only a
section of the growing season with warmer temperatures and half to full-grown vegetation. The
importance of ER is likely to be different when considering the full annual balance (*e.g.,* Hashemi
et al. 2021). While our data represent only the growing season, a similar relationship has also been
found between the annual NEE and LAI at a tundra site with a mixture of wet and dry tundra in
northeastern Europe (Marushchak et al. 2013), in a multi-site EC study in Alaskan tundra



(McFadden et al. 2003), in Canadian low arctic tundra wetlands (Lafleur et al. 2012), and across
tundra sites (Street et al. 2007; Shaver et al. 2007).

The magnitude of $Pg_{800}$ and $NEE_{800}$ in the fen and meadow plots of this study were

similar to the maximum Pg and NEE found in tundra wetland in Seida in northeastern Europe
(Marushchak et al. 2013), at low tundra wetland sites in eastern Canada (Lafleur et al. 2012), and at
a wetland-dominated but more continental site (with an equal growing season length) in
northeastern Siberia (van der Molen et al. 2007). The vegetation and $Pg_{800}$ of lichen tundra and
dwarf-shrub tundra in our study resembled those observed within the polygon rim habitat of the
polygon tundra in the Lena River delta, while those of meadow, dry fen, and wet fen resembled the
wet polygon center habitats (Eckhardt et al. 2019). In our study, the variation of ecosystem
respiration resulted from the variation in vascular plant LAI, soil organic content, and water
saturation: the highest ER occurred in mineral soil meadow with high LAI suggesting substantial
autotrophic respiration and likely deep rooting and large root biomass contributing to the ecosystem
respiration (Fig. 4). In wetlands, respiration may be attenuated by the soil water saturation.

Our chamber-based estimate of the average $CH_4$ flux within the 35.8 $km^2$ upscaling area

was 0.05 mmol $m^{-2}$ $h^{-1}$, which is close to 0.04 mmol $m^{-2}$ $h^{-1}$ obtained by Tuovinen et al. (2019) who
combined EC data with footprint modeling to statistically determine LCT group-specific $CH_4$
fluxes. Within this upscaling area, 28% of the area emitted $CH_4$, while the other habitats either
consumed atmospheric $CH_4$ (lichen tundra including barrens, coverage 26%) or were close to
neutral relative to the atmosphere (Figs. 4, 5, Table 3). The wettest spots were the sites having the
highest $CH_4$ emissions (Fig. 4). We observed no clear relationship between vegetation and $CH_4$ flux
in plot level, which could partly be due to the small size of data. At a LCT level, high LAI and high
$CH_4$ emissions co-occurred if WT was high enough (Fig. 3). The sites showing the highest
emissions had a high soil organic matter content, an indication of slow decomposition in anoxic
conditions, and we also found that the eroded bare-peat surface of dry fen and the disturbed vehicle



tracks had high $CH_4$ emissions. In the case of eroding surfaces, gas efflux may be enhanced by
transport pathways emerging from changes in soil structure. Wet depressions, like the vehicle tracks
in this study, have in turn been found to have high $CH_4$ emissions relative to their surroundings in
permafrost, which results from the abundance of graminoids producing easily degradable litter
compared to dwarf-shrubs, and the potentially increasing nutrients from seasonal permafrost
degradation (*e.g.,* Bubier et al. 1995, McCalley et al. 2014, Wickland et al. 2020). All in all, our
data encourage applying indicators of wetness together with vegetation parameters as a means of
$CH_4$ flux upscaling in tundra environment. While the topographic wetness index in general was a
reasonable surrogate for WT, distinguishing the dry and wet soils, erroneous TWI values were
estimated for the streamside meadow, possibly due to insufficient locational accuracy, because the
plots were located right next to the stream, but on an elevated bank.

The recognition of $CH_4$ consuming tundra habitats is important for accurately estimating

the net $CH_4$ balance of tundra. The substantial uptake of atmospheric $CH_4$ by lichen tundra (here a
mixture of bare ground and sparse vegetation) in Tiksi was inferred by Tuovinen et al. (2019) based
on a source allocation analysis of EC data: the average flux of the consuming area was estimated at
-0.03 mmol $m^{-2}$ $h^{-1}$, which corresponds to -21.6% of the total upscaled $CH_4$ flux. In this study, the
average growing season $CH_4$ uptake was -0.02 mmol $m^{-2}$ $h^{-1}$ in the lichen tundra plots and an order
of magnitude lower in graminoid tundra, dwarf-shrub tundra, and bog. Our upscaling exercise
resulted in a $CH_4$ sink that counterbalanced about -10% of the $CH_4$ emission, which likely is an
underestimate due to an overestimation of the emissions from the wet fens. High consumption of
atmospheric $CH_4$ in barrens is associated with the high affinity methanotrophs (Jørgensen et al.
2014; Lau et al. 2015; D'Imperio et al. 2017, St Pierre et al. 2019). For instance, on Disko Island,
Greenland, which consists of similar land cover types to Tiksi, uptake of $CH_4$ by bare ground was -
0.005–0.01 mmol $m^{-2}$ $h^{-1}$ during the growing season, while a mean flux of -0.003--0.004 mmol $m^{-2}$
$h^{-1}$ was observed in dry tundra heath (D'Imperio et al. 2017). These consumption rates associated



with tundra barrens and high-affinity methanotrophs can be high relative to consumption rates
measured on north-boreal forest soils (for instance, -0.01 mmol m$^{-2}$h$^{-1}$, Lohila et al. 2016).

**5 Conclusions**
Our results provide new observations of carbon exchange for the prostrate dwarf shrub tundra sub-
zone, which covers an area of 2.3 million km$^2$ of the Arctic (Walker 2000). Graminoid vegetation
favored the wet and moist habitats, such as wet fens and the streamside meadow, which were
characterized by large $CO_2$ uptake and $CH_4$ emissions. The heterogeneity of landscape and the
related large spatial variability of $CO_2$ and $CH_4$ fluxes observed in this study encourage to monitor
the Arctic sites for changes in habitat type distribution. Such changes can include the forming of
meadows and appearance of new vegetation communities, such as erect shrubs, that benefit of
warming-induced changes in thaw depth and soil wetness. The spatial extrapolation based on a
small number of measurement points involves inherent uncertainty but still allowed us to identify
key relationships between $CO_2$ and $CH_4$ fluxes and vegetation and moisture features, which can be
utilized in more robust upscaling experiments that make use of EC measurements.

*Data availability*. The flux data used in this study can be accessed via the Zenodo data repository:
Juutinen, Sari. (2022). Dataset for a manuscript entitled Variation in CO2 and CH4 Fluxes Among
Land Cover Types in Heterogeneous Arctic Tundra in Northeastern Siberia [Data set]. Zenodo.
https://doi.org/10.5281/zenodo.5825705







*Author contributions*

TL, MA, and SJ designed the study. TL, MA, and AM took care of the overall site governance and maintenance. VI, ML, TL, JM, JN, EV, TL, TV, and MA conceived the field measurements of $CO_2$ and $CH_4$, vegetation, and environmental variables. In addition, ML calculated green chromatic coordinates, and MA and J-PT postprocessed the EC data and J-PT modeled the footprint and estimated footprint LCT fractions. AR and TV processed and modelled the landcover data and estimated TWI and NDVI for the plots and area. SJ compiled the chamber flux data and conducted the data analyses and spatial extrapolations and wrote the manuscript with contributions from all co-authors.

*Competing interests*

The authors declare that they have no conflict of interest.

*Acknowledgements*

We thank G. Chumachenko, O. Dmitrieva, and E. Volkov at the Tiksi Observatory and the Yakutian Hydrometeorological Service for their kind assistance in carrying out and organizing the field campaigns and Lauri Rosenius for assistance in the field work. This study was financially supported by the Academy of Finland, projects "Greenhouse gas, aerosol and albedo variations in the changing Arctic" (project no. 269095), "Carbon balance under changing processes of Arctic and subarctic cryosphere" (project no. 285630), "Constraining uncertainties in the permafrost-climate feedback" (project no. 291736) and "Carbon dynamics across Arctic landscape gradients: past, present and future" (project no. 296888); the European Commission, FP7 project "Changing permafrost in the Arctic and its global effects in the 21st century (PAGE21, project no. 282700)"; and the Nordic Council of Ministers, DEFROST Nordic Centre of Excellence within NordForsk.



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





**Appendix**


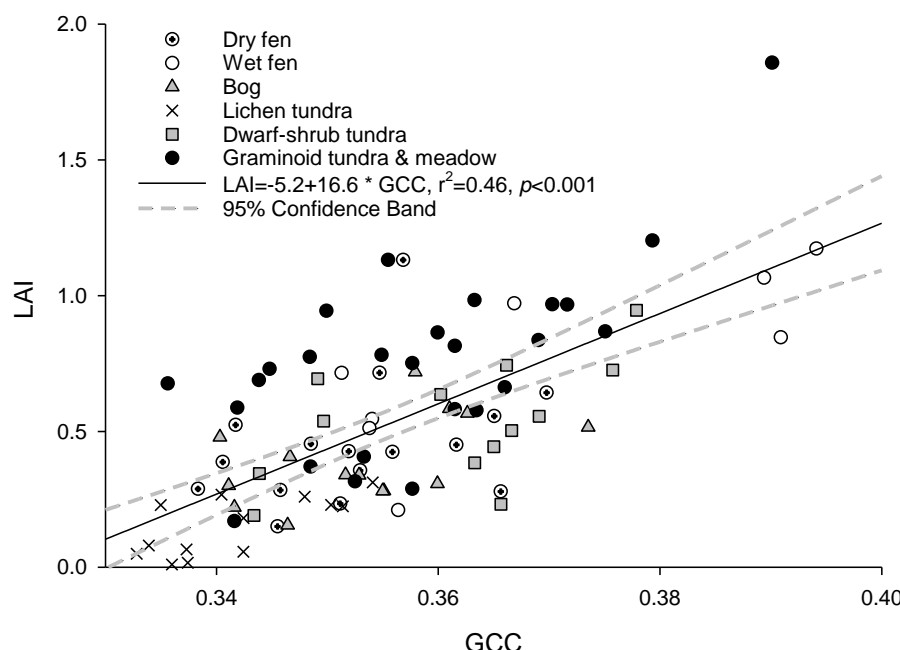




Appendix Figure 1. Relationship between GCC and vascular plant LAI in the harvested plots. LCTs
are indicated with symbols. In the LCT-specific regressions (not shown), the coefficient of
determination ($R^2_{adj.}$) was lowest for dry fen (0.06) and highest for wet fen (0.54). Regression
slopes varied from 8.3 for dry fen to 17.8 for the combined graminoid tundra and meadow LC