# Peer review of "Variation in CO2 and CH4 Fluxes Among Land Cover Types in Heterogeneous Arctic Tundra"

_Biogeosciences, 2022_

## Author Comment (AC1)

**Author Comments to Referee 1**
Juutinen et al., Variation in $CO_2$ and $CH_4$ Fluxes Among Land Cover Types in Heterogeneous Arctic Tundra in Northeastern Siberia

We thank both reviewers for their time, thorough reviews, and valuable comments. We have edited the manuscript according to the comments and suggestions. Following the suggestion by Reviewer 1, we checked the $CH_4$ consumption data from the barren tundra. We found an error in the case of the most negative $CH_4$ flux and recalculated the flux in two other cases. The effect on the mean seasonal value was small and did not change the landscape-level $CH_4$ estimate much. The substantial $CH_4$ consumption rate in the barren tundra is real and was detected also by the EC-measurements (this study and Tuovinen et al. 2019). We revised the text following the suggestions and emphasized the role of barren tundra as $CH_4$ sink in this landscape. In Tiksi, the largest consumption of atmospheric $CH_4$ occurs in barren that is composed of sand and small rocks. Even though the $CH_4$ consumption rates were large, the pattern and values agree with those measured in a few other circumpolar polar deserts and barrens, which we show in a review table (Table 4). To emphasize the $CH_4$ sink function found in this landscape, we appended the introduction, material and methods, results and discussion at suitable places. Reviewer 2 pointed out that showing results of temporal $CH_4$ flux dynamics did not serve to answer the questions set in the study. We have streamlined the text accordingly and edited Figures 5 and 6. Both reviewers criticized our sloppy utilization of the DCA analysis results, which we acknowledge. We think that the DCA summarizes many features of the landscape and we are opting to include it. Nevertheless, we have put effort to linking the DCA graph and the text. Please find below our point-by-point answers to each comment (**AR:** blue type).

Sincerely,

Sari Juutinen on behalf of all authors
* * *
**Reviewer 1**

General and specific comments

The manuscript "Variation in CO2 and CH4 fluxes among landcover types in heterogeneous Arctic tundra in Northeastern Siberia" by Juutinen et al. presents several years of CO2 and CH4 flux data, measured both with manual chambers as well as the eddy covariance technique. The authors combine their flux measurements with detailed investigations of site vegetation characteristics and site meteorological data, measured at an Arctic tundra site in Siberia.

This is an important study because it highlights the difficulties in determining C emissions from these heterogenous ecosystems. The study is set in an understudied region in terms of C exchange, and considering how challenging measurements in these remote regions are, I highly value the multi-year data series that are presented here. Further, there are only a few studies that report C fluxes measured with different techniques simultaneously, as is done here, and studies such as this are very much needed to improve our ability to constrain the high-latitude C budget. I also

appreciate the detailed and thorough vegetation analyses performed in this study to accurately determine LAI and linking vegetation characteristics to fluxes.

I have a couple of comments that I encourage the authors to address before publication.

1) I suggest adding a few sentences discussing the possible reasons for the observed differences between manual chamber and eddy covariance estimates observed here

AR: The two estimates give similar trends. Differences in the magnitudes most likely arise from the small number of the chamber measurement plots and accuracy of the LCT map. We have speculated this in the discussion.

2) Please add a short explanation of high- vs. low affinity methane oxidation as well as of barrens for the general benefit of the reader (lines 78-80). Barren tundra surfaces can be quite different from each other (rocky surface with thin or absent organic layer/polar deserts, or eroding surfaces in more organic-rich areas, peatlands). Would be good to know which type the authors refer to here, and if $CH_4$ uptake occurs from all barren surfaces or some ecosystem types in particular. Similarly with high-affinity methane oxidation: $CH_4$ oxidation in high vs low $CH_4$ environments (low- vs. high affinity methanotrophy) are important concepts for this study looking at contributions from wet vs dry tundra, so they should be adequately addressed in the introduction if mentioned.

AR: This is an important comment. We have improved the site description and the different $CH_4$ consumption concepts and the importance of the consumption overall in the introduction and elsewhere in the text.

3) The measured $CH_4$ uptake rates seem rather high, especially some of the maximum values presented in Fig. 5 for lichen tundra. I consider the observed large contribution as a $CH_4$ sink of this landcover class to the regional $CH_4$ balance an important finding and agree it is important to highlight this in this study as the authors have done. However, I am skeptical of these very large flux rates that seem to be one order of magnitude larger than what has been reported previously (references below). Since this is a potentially important message of the manuscript, I would suggest the authors double check the slopes used for calculating manual chamber fluxes and start point gas concentrations, and afterwards re-evaluate if the reported 10% offset of $CH_4$ emissions by $CH_4$ consumption is accurate.

Looking at Fig. 5, the maximum $CH_4$ uptake goes as low as -0.1 mmol $CH_4$ m-2 h-1. If my conversion is correct, this corresponds to -39 mg m-2 d-1. This would seem like an unreasonable large flux to me, considering diffusion constraints of atmospheric $CH_4$ into soils. I recommend the authors double-check at least these large uptake rates, as they may substantially distort the mean.

- Are these manual chamber measurements (flux calculation based on only a few data points and lower accuracy when measuring with GC) or were these fluxes measured with the LGR?

- what was the initial concentration at the start of the measurement/starting point of the selected slope? Did the authors check these concentrations were close to ambient? Otherwise, a starting concentration above ambient after chamber placements may not yield realistic flux estimates.

- what was the minimum number of points included, e.g. for manual sampling with 4 time points, were always for points used for determining the flux or even less?

- Reported EC values are in the same range. Is the closed-path eddy covariance instrument that was used reliable for low concentration (below ambient) measurements, or do these concentrations have to be taken with a grain of salt? Any issues with instrument noise for the low end of fluxes?

Compared to CH4 uptake reported from northern soils (Arctic + boreal) these values would appear one order of magnitude larger than could be expected. In lines 499-502 the authors compare their fluxes (mean: 0.02 mmol m-2 h-1, max 0.1 mmol m-2 h-1) to CH4 uptake rates determined at similar sites which were about one order of magnitude smaller (0.005-0.01 mmol from bare ground, 0.003-0.004 mmol m-2 h-1, ref D-Imperio et al. 2017), and are in the range of what has been reported from Arctic-boreal synthesis studies on CH4 fluxes from a large number of sites. I suggest comparing with some of these studies, for example the following references:

Kuhn, M. A., Varner, R. K., Bastviken, D., Crill, P., MacIntyre, S., Turetsky, M., ... & Olefeldt, D. (2021). BAWLD-CH 4: A Comprehensive Dataset of Methane Fluxes from Boreal and Arctic Ecosystems. Earth System Science Data Discussions, 1-56.

Bartlett, K. B., & Harriss, R. C. (1993). Review and assessment of methane emissions from wetlands. Chemosphere, 26(1-4), 261-320.

E.g., Bartlett&Harriss report that CH4 uptake from these ecosystems is generally < -2 mg CH m-2 d-1 on average, and the more recent synthesis by Kuhn et al. report uptake in the range of -1.1 - -0.17 mg CH4 m-2 d-1.

**AR**: The large consumption rates occurred in schist-like rocky surface with negligible organic matter (Fig 1 below). We checked the outlier data point (the most negative value) and found a typing error. In two other cases we recalculated the flux using a longer closure period and the effect of initial high flux after the closure evened out. We opted for the more conservative estimate in these cases. Most of the data were measured using the LGR analyzer, and we show below (and in the ms appendix) examples of chamber concentration time series (Fig. 2 below). The precision of the instrument is sufficient to detect small fluxes, and there are more than 60 data points per a closure. The starting concentrations were checked and confirmed to be ambient. The mean flux for the barren decreased after recalculating the three highest fluxes, but the consumption rate of the barren surface remains high. Consequently, the estimate of total $CH_4$ consumption relative to the emissions in the landscape decreased, and the resulting shares of $CH_4$ flux in barren and all the consuming LCTs were 6% and 8%, respectively, of the emissions. Tables and figures are corrected accordingly. Overall, the large consumption of $CH_4$ was evident and was also detected by the EC measurements. That is shown in Fig. 7f, and also earlier by Tuovinen et al. (2019). The sensitivity and precision of the EC instrument are sufficient to detect the small downward $CH_4$ fluxes, especially when averaged over a longer time period. Fig. 5 in Tuovinen et al. (2019) shows the $CH_4$ fluxes in different wind direction sectors.

We compiled, however, reference data into a table now included in the text (see below and Table 4) and enhanced the discussion. In this review, we focused on similar dry tundra sites, including barren and polar desert sites and relevant data from the BALDW synthesis (Kuhn et al. 2021), which are from the study of Emmerton et al. (2014), incl. five sites on Ellesmere Island. The table indicates, as

already known, that the barren surfaces with limited organic matter and energy supply for the microbes tend to have large $CH_4$ consumption rates.

[Figure]

Fig. 1. (Ms Fig. A1) Examples of the barren (left) and lichen tundra (right) plots with close views. Vegetation consists of, for instance, lichens, *Dryas octopetala,* and dwarf shrubs *Vaccinium vitis-ideae.*

[Figure]

Fig. 2 (ms Fig. A2). Examples of gas concentration variations in chambers measured using the LGR analyzer. The examples represent lichen tundra, barren, and wet fen.

Table 1 (ms Table 4). Review of reported consumption rates of atmospheric $CH_4$ in dry mineral soil tundra incl. this study.

| Location | Habitat type | Mean | Min | Max | Reference |
|---|---|---|---|---|---|
| | | ($\mu mol\ m^{-2}h^{-1}$) | | | |
| Narsarsuaq, Greenland | low elevation heath vegetation | -1.2 | -4.0 | -0.2 | St Pierre et al. 2019 |
| Narsarsuaq, Greenland | high elevation heath vegetation | -2.6 | -11.9 | 3.6 | St Pierre et al. 2019 |
| Disko Island, Greenland | low elevation heath vegetation | -3.8 | -12.1 | -1.1 | St Pierre et al. 2019 |
| Disko Island, Greenland | high elevation heath vegetation | -3.5 | -12.1 | -1.3 | St Pierre et al. 2019 |
| Tierra del Fuego, Argentina | alpine tundra | 0.5 | -16.6 | 10.3 | Sá et al. 2019 |
| Disko Island, Greenland | dry tundra heath[1] | -4.0 | -4.4 | -2.5 | D'Imperio et al. 2017 |
| Disko Island, Greenland | bare ground[1] | -9.0 | -15.0 | -3.8 | D'Imperio et al. 2017 |
| Disko Island, Greenland | *Betula nana* and *Salix* sp. heath | -4.0 | | | Christiansen et al. 2014 |
| Axel Heiberg Island, CA | vegetated ice-wedge polygon | | -2.7 | -0.3 | Lau et al. 2015 |
| Lake Hazen, Ellesmere I., CA | polar desert[2] | -3.6 | -7.0 | 0.0 | Emmerton et al. 2014 |
| Zackenberg Valley, Greenland | moist tundra | -3.1 | -7.0 | -2.0 | Jørgensen et al. 2014 |
| Zackenberg Valley, Greenland | dry tundra & barren ground | -7.0 | -16.0 | -4.0 | Jørgensen et al. 2015 |
| Zackenberg Valley, Greenland | tundra heath | -1.3 | -6.0 | 0.0 | Christensen et al. 2000 |
| Okse Bay, Ellesmere I., CA | polar desert[3] | -0.5 | | | Brummel et al. 2014 |
| Petterson R., Ellesmere I., CA | polar desert[3] | -0.04 | | | Brummel et al. 2014 |
| Dome, Ellesmere I., CA | polar desert[3] | -0.5 | | | Brummel et al. 2014 |

| | | | | | |
|---|---|---|---|---|---|
| BAWLD-CH$_4$ Synthesis | dry tundra | | -2.9 | 5.2 | Kuhn et al. 2021 |
| BAWLD-CH$_4$ Synthesis | boreal forest | | -2.6 | -0.5 | Kuhn et al. 2021 |
| Tiksi, RU | lichen tundra mean | -11.3 | -57.9 | -0.4 | This study |
| Tiksi, RU | barren | -18.1 | -57.9 | -3.0 | This study |
| Tiksi, RU | vegetated | -6.0 | -34.7 | -0.4 | This study |
| Tiksi, RU | meadow | -1.0 | -21.1 | 24.5 | This study |
| Tiksi, RU | dwarf-shrub tundra | -0.2 | -2.9 | 20.3 | This study |
| Tiksi, RU | bog | -2.1 | -14.8 | 6.6 | This study |

1) mean estimated from a figure, 2) min and max estimated from a figure, 3) one-three day

measurement

Line edits

Introduction

L62: and warming?

**AR**: text edited

L78: add reference. Also, useful to add that dry tundra is often reported as CH4 neutral, not necessarily as a small sink even. A recent reference that the authors may find useful: Kuhn, M. A., Varner, R. K., Bastviken, D., Crill, P., MacIntyre, S., Turetsky, M., ... & Olefeldt, D. (2021). BAWLD-CH 4: A Comprehensive Dataset of Methane Fluxes from Boreal and Arctic Ecosystems. Earth System Science Data Discussions, 1-56.

**AR**: Text edited accordingly

L78-80: a short explanation of tundra barrens and high-affinity methane oxidizers would be useful in this context (see comment above).

**AR**: Text changed as suggested

L87-88: Please be more specific – biased towards what? Does this mean in heterogeneous environments estimates are biased towards emissions? Or biased in that sense that an integrated flux does not yield sufficient information on sink/source behaviour of individual landcover types?

**AR**: The EC measurement may bias the flux estimate if the surface source/sink distribution is heterogeneous. This results from the fact that, even though EC integrates spatially, the spatial integration involves non-uniform weighting of the upwind surface elements (flux footprint).

Furthermore, the EC sampling varies temporally depending on meteorological conditions (footprint climatology), which affects the resulting flux statistics. Fluxes are not systematically biased towards either emission or uptake; the bias depends on the distribution of footprints vs that of sources/sinks (see Tuovinen et al., 2019). We rephrased the text.

Methods

L110: delete "normal"

**AR:** That's replaced by climate normal

L113: soil organic matter content? Additionally, please provide some information of organic layer thickness at the site in the methods text, and refer to Table 1. Based on the reported low OM content, lichen patches are located exclusively on mineral soil with very thin or no organic layer? Do the authors have any information on the lichen species that could be added?

**AR**: The most abundant lichen genera were added into the Table 1. Those are *Thamnolia, Flavocetraria,* and *Alectoria*. Information about the organic layer depths added in the text: it is negligible in lichen tundra (and barren), a few cm in the dwarf-shrub tundra, meadow, and graminoid tundra, and at least 30–40 cm in bog, dry fen, and wet fen (Mikola et al. 2018)

L170: please add specifics of vials used for storage as well as type of GC (manufacturers, volume, tested for gas tightness during storage, how long were samples stored before analysis?)

**AR**: The following specifications are included in the text now. "four air samples were taken from the chambers using syringes (250 mL). The sample was pumped into a 50 mL glass vial with a rubber septum. The vial was purged with the sample and the vial was over-pressurized with the last 10 mL of the sample. Methane concentrations in samples were determined at the laboratory of the Voeikov Main Geophysical Observatory using a TSVET 500-M gas chromatograph (Chromatek, Ru) with a flame ionization detector. Each measurement was accompanied by calibration using standard gas mixtures with the NOAA2004 scale. The samples were analyzed within a month. The vials were tested prior the measurements: after two weeks, the samples were still over-pressurized and $CH_4$ concentrations were within ±3 ppb the concentration in the standard gas".

L173: Was the 5-minute enclosure time applied to all surfaces, and was this enclosure time sufficient to accurately determine slope for low emitting (or uptake) sites?

**AR**: It was enough. The analyzer has a high precision.

L178: What about non-linearity due to PAR for CO2 measured with transparent chambers?

**AR**: Data were screened for variation in PPFD and rejected if the variation exceeded 100 $\mu$mol m$^{-2}$s$^{-1}$. Now mentioned in the text.

L174-178: Where there some general rules applied as to how many points were usually discarded at the beginning of each measurement, and how many points were included for flux calculation? How was the quality of fluxes assured (R2, RSME, other)? Please add some specifics.

AR. We added specifics of the flux determination in the text: The first data points were generally neglected when determining the slope of concentration change over time and cases with linear concentration change had coefficient of determination ($R^2$) > 0.9. No change in concentration meant zero flux. There were a few ebullition cases at the vehicle track measurement points that had only sparse or no vegetation cover. When determining NEE using the transparent chamber, the data were screened for variation in PPFD and rejected if the variation exceeded 100 $\mu$mol m$^{-2}$ s$^{-1}$.

L262: why were different classes for graminoid tundra applied to CO2 and CH4 and not the same for both gases?

**AR:** That was based on assumed flux rate similarity based on moisture and vegetation characteristics.

Results

L297-298: Check sentence structure.

**AR**: Right, the text edited.

L330: This is indeed quite large as a mean flux for atmospheric CH4 consumption. Please see specific comments above.

**AR**: See our reply above. Even though the largest negative flux data point was erroneous, and we applied more conservative data selection for the other two large value, the mean is still large. The large sink is supported by the EC observations, and the EC estimate for the northern sector shows actually larger consumption of CH$_4$ than our chamber-based estimate.

L367-398: It would be interesting to see the time series of CO2 and Ch4 fluxes measured with the eddy covariance technique, instead of just mean numbers based on wind sector contribution. That way the reader would get a better overview of the timing of high/low fluxes or possible peaks that would help interpret the data and help understand discrepancy between chamber and EC data. This time series could be in shown as supplementary in case the authors are tight for space in the main text.

AR: We compared data measured in the same time window. We refer here to Figure 5 in Tuovinen et al. (2019) that shows the 30-min CH4 fluxes as a function of wind direction. The range of these fluxes is about -0.045 to 0.18 mmol/m2/h, the limits corresponding to the barren and wet fen values measured with the chambers.

Discussion:

L422: Do the authors mean 9% ? Early they state 10%.

**AR**: We checked the consistency and replaced with a revised value (that is 8% of the emissions)

L430-431: their high OM content is already mentioned in line 426.

**AR**: We edited the sentence

L441: soil organic matter?

AR: The sentence was edited (l. 462)

Figures and tables:

Fig. 2: add information on landcover class (wet fen, dry tundra) in figure panels c), d) and e) instead of just in the figure caption. Add info on missing thaw depths measurements (panel f) for some landcover types (e.g., too rocky under lichen cover) in figure caption.

AR: Changed as suggested

Fig. 3: please add percent explanatory power to each component axis (xx%). The DCA is very much dominated by the high CH4 fluxes from wetlands. The authors may want to consider adding a second panel to this figure, where they provide DCA only for low-emitting and uptake sites, to identify the influence of environmental settings on low fluxes. Also, is there a reason why soil temperature was not included in the figure?

AR: As suggested the percentages explained are now shown in the axis titles. We think that it is unnecessary to show only the dry tundra plots as a DCA plot, because it is already evident from the current plot, i.e. minimal vegetation and dry conditions. Soil temperature was not comprehesively measured across the plots and thus we opted not to use it.

Fig. 5: Please see my comments above regarding large uptake in lichen tundra. Additionally, consider colouring the fluxes by measurement year.

AR: The figure is revised according to the data changes. The data distribution is explained in Table 2 and therefore the symbols are unchanged.

Fig. 6: Symbols for vehicle track and bar

AR: The figure has been edited and those surfaces are included in the current version.

---

## Author Comment (AC2)

**Author Comments to Referee 2**
Juutinen et al., Variation in $CO_2$ and $CH_4$ Fluxes Among Land Cover Types in Heterogeneous Arctic Tundra in Northeastern Siberia

We thank both reviewers for their time, thorough reviews, and valuable comments. We have edited the manuscript according to the comments and suggestions. Following the suggestion by Reviewer 1, we checked the $CH_4$ consumption data from the barren tundra. We found an error in the case of the most negative $CH_4$ flux and recalculated the flux in two other cases. The effect on the mean seasonal value was small and did not change the landscape-level $CH_4$ estimate much. The substantial $CH_4$ consumption rate in the barren tundra is real and was detected also by the EC-measurements (this study and Tuovinen et al. 2019). We revised the text following the suggestions and emphasized the role of barren tundra as $CH_4$ sink in this landscape. In Tiksi, the largest consumption of atmospheric $CH_4$ occurs in barren that is composed of sand and small rocks. Even though the $CH_4$ consumption rates were large, the pattern and values agree with those measured in a few other circumpolar polar deserts and barrens, which we show in a review table (Table 4). To emphasize the $CH_4$ sink function found in this landscape, we appended the introduction, material and methods, results and discussion at suitable places. Reviewer 2 pointed out that showing results of temporal $CH_4$ flux dynamics did not serve to answer the questions set in the study. We have streamlined the text accordingly and edited Figures 5 and 6. Both reviewers criticized our sloppy utilization of the DCA analysis results, which we acknowledge. We think that the DCA summarizes many features of the landscape and we are opting to include it. Nevertheless, we have put effort to linking the DCA graph and the text. Please find below our point-by-point answers to each comment (**AR:** blue type).

Sincerely,

Sari Juutinen on behalf of all authors

\*\*\*\*\*\*\*\*\*\*\*\*\*\*\*\*\*\*\*\*\*\*\*\*\*\*\*\*

**Reviewer 2**

Summary:

For their study, the authors performed closed chamber measurements of CO2 (in 2014) and CH4 (between 2012 and 2019) fluxes in different land cover types (LCTs) in Northeastern Siberia during the growing season along with supporting meteorological measurements. Upscaling of the chamber data and comparison with eddy covariance (EC) measurements revealed the importance to distinguish between different land cover types when estimating tundra C exchange on a larger spatial scale: Mainly driven by differences in vegetation coverage and soil wetness, tundra wetlands contributed disproportionately much to the total CO2 uptake and CH4 emission regarding their spatial extent. Drier tundra landcover types instead offset the CH4 emissions through significant consumption of CH4.

Major comments:

The questions addressed in the study are well within the scope of BG. The study does not really comprise any new ideas or concepts, however publishing greenhouse gas flux data and additional measurements from the still data-scarce Arctic region is valuable in itself. From my point of view (and as the authors state themselves) the small number of replicates per LCT does not allow for a precise quantitative evaluation of greenhouse gas emission depending on the LCT. I expect that assuming that a single plot per LCT (as for example in 2014 for bog and dwarf-shrub tundra, see Table 2) is representative for the whole LCT, might introduce high uncertainty into the upscaled data product. For example different microtopography types within a bog (small hummocks, hollows,…) might already show very different exchange rates of greenhouse gases. The study clearly focusses on the spatial aspect, however, many more temporal replicates were performed. The design of the measurements therefore does not match the aim of the analyses very well. Regarding this issue it is nearly surprising to me, how well the upscaled chamber measurements match the EC measurements (at least from a qualitative point of view) (Figure 7). The main conclusion that different land cover types should be distinguished for upscaling is not new but the proof of its importance, given in the paper, is still useful also regarding possible future changes in the distribution of different LCTs due to climate change.

A new aspect is added to the study by the multivariate analysis that investigates the relationship between gas fluxes and environmental variables. However, this analysis seems a bit redundant to me in this context because it does not add any information to the results or conclusions presented in the paper. Furthermore, the DCA ordination diagram (Figure 3) is only described in a rather technical manner. In my opinion the multivariate analysis should either be removed from the paper or it should be described, analyzed and interpreted in more detail.

In general more information is included in the manuscript than is needed to answer the research questions (e.g. also the temporal differences between CH4 fluxes within the growing season). This sometimes makes the manuscript hard to follow. In my opinion it would be better to focus on the data that is relevant for the study aim.

Throughout the manuscript words are sometimes written out although an abbreviation had been introduced earlier. Adding an overview table that contains all the abbreviations would be helpful also because there are quite some abbreviations used in the manuscript.

**AR**: We thank for the comments and suggestions. We streamlined the text to focus on the spatial aspects. We admit the low number of spatial samples that is a result of compromising for determining the light response of net exchange of $CO_2$, which requires temporal replication. Also, the measurement campaigns had time constraints. We specified in the text that 'bog' means here the drier tundra peatland that is distinguished from the fens by more abundant dwarf shrubs and lower WT position. In this case, there is no pool and hummock variation. This does not remove the fact that replication per LCT is low. We decided to show the temporal aspect of CH4 fluxes only to reveal the data behind the mean values. However, we redrew Figures 5 and 6 to focus on the spatial aspect and edited the text consequently. In addition, we have elaborated the interpretation of the DCA-analysis in results and discussions. The usage of abbreviations was checked throughout the text, but no table was included, because their number is moderate.

Minor comments:

l. 78: The word "act" is missing and "-s"

AR: Revised

l. 86: I don't understand the meaning of the word "enhances" in this context

AR: "Enhances" replaced by "improves"

ll. 86, 87: if only the eddy covariance method is meant with "micrometeorological measurements", I would mention this explicitly.

AR: Changed as suggested

l. 96: In ll. 87, 88 it is mentioned that flux estimates using the eddy covariance technique might be biased in a highly heterogeneous environment like the study area. Is it then reasonable to compare the chamber measurements to the eddy covariance measurements to assess the spatial representativeness of the chamber method? It is certainly helpful to compare chamber and EC measurements but the way the reasoning is expressed here it seems a bit contradictory. Maybe you could just rephrase your reason for comparing the chamber fluxes with eddy covariance measurements.

AR: The possible EC bias is taken into account in the EC vs chamber comparison by weighting the LCT-specific chamber-based fluxes by LCT proportions (Fig. 7). These proportions represent the relative contribution of each LCT as estimated from the EC footprint climatology and LCT map, and thus this weighting improves the comparability of the chamber- and EC-based fluxes. We rephrased the text.

l. 106: At several point in the manuscript, when referring to a figure, I would add the relevant part of the figure to the reference. For example in this line I would refer explicitly to Figure 1a instead of just Figure 1.

AR: We have edited the figure references throughout the text.

l. 117: I cannot see this from Figure 1 and would therefore only refer to Table 1.

AR: Done

l. 123: I would also refer to Figure 1 d-h here.

AR: Done

l. 157: "…over 5 °C…" – is that the definition of the growing season?

AR: Not necessary in the Arctic, but that value was used just to indicate conditions.

ll. 176 – 179: Since the analyses are based on little replicates it would be interesting, how many measurements had to be discarded. Maybe this information could be added to Table 2, if the numbers do not already give only the valid flux measurements.

AR: Only valid data included

ll. 229 – 238: How exactly was the "light response of Pg and NEE" determined? How exactly did you determine the value of Pgmax and Pg800?

**AR:** Text has been edited.

l. 238: What do you mean with "collar means"? Are these temporal means over all the measurements performed at one collar?

**AR**: Mean of observations per each collar, temporal means. Text has been edited.

l. 254: A bracket is missing after "…360°"

**AR**: Corrected

l. 275: I would refer only to Figure 2b here.

**AR**: Corrected

l. 276: "2011-2019"

**AR**: Corrected

l. 282: The reference should be to Figure 2 c-d.

**AR**: Corrected

l. 291: a "T" for temperature is missing after "…soil surface…"

**AR**: Corrected

l. 297: The sentence structure does not make sense.

**AR**: Corrected

ll. 301, 302: Why is the strong correlation of ER with axis 2 not mentioned?

**AR**: Not very strong, but it is included in the text now

l. 313: What is the meaning of these Eigenvalues?

**AR**: The eigenvalue is a measure of the strength of an axis, the amount of variation along an axis, and ideally the importance of an ecological gradient. We report in the text the variances explained by each axis in the vegetation data. Eigenvalues can be removed.

ll. 313, 314: I would rather add the information that "…axis 1 and 2 explain cumulatively 63% of the variation…" to the main text than keeping it in the figure caption.

**AR**: Edited as suggested and values are removed from the caption and the axis statistics are in the text.

l. 335: According to Figure 4 there is no significant linear relationship between CH4 fluxes and WT...

**AR:** Not linear but $CH_4$ flux was related to WT. Wording changed and correlation replaced by related.

l. 345: Is the standard error the same as standard deviation? In Figure 6 standard deviation is used and in Table 3, standard error.

AR: No, it is different. Both were edited and it is standard error in the table and figures.

l. 364: the "4" in "CH4" should be made into a subscript

AR: Corrected

ll. 370, 371: I would say "…comprised…of…" or "…contributed…to…"

AR: Edited

l. 377: I would explicitly refer to Figure 7 b-d.

AR: Edited

ll. 379, 380: Which wind sector do the percentages refer to?

AR: Those are averages. Text edited accordingly.

l. 382: I would refer to Figure 7f.

AR: Edited

l. 392: "…exchange of CO2, photosynthesis, and CH4 flux,…"

AR: Edited, now stands "…NEE, Pg, and $CH_4$ emissions…"

l. 401: "…wind direction sectors (a)),…". Which years are included for Figure 7 f)? Only 2014 or all years of CH4 flux measurements?

AR: The legend edited, and it is also mentioned that only year 2014 data were used.

l. 409: What are the "collar-specific estimates"?

AR: Estimates were calculated for each collar. Text edited.

l. 418: Does the "bog" not count as a wetland type?

AR: Right, reworded in the text. Graminoid types had large CO2 uptake capacity while the wet graminoid types, i.e., fens, emitted CH4.

l. 422: "%" is missing. Is it 9 or 10%? At other points of the manuscript you write that it is 10%.

AR: The value is recalculated and corrected; 8%.

l. 435: "not" instead of "neither"

AR: Corrected

l. 473: Better to also refer to Figure 6.

AR: Edited

l. 475: I cannot see this from Figure 3.

AR: Text edited but still citing the Fig. 3.

l. 476: How was the soil organic matter content inferred? The data is not shown anywhere.

AR: Right, the soil OM data was published in Mikola et al. (2018), now included in the text.

l. 497: Why do you expect "an overestimation of the emissions from the wet fens"?

AR: Reworded and the fact that EC saw even higher consumption for the northern sector is included as a more relevant fact here.

Comments to Figures and Tables:

Figure 1b):

I would be nice to either give a closer view of the map so that it can be seen in which LCTs the chamber measurements were performed or (which would be even nicer) mark the EC footprint (impact area) on the map. Is the "stony" LCT the same that is referred to as "barren" in the text? It would be helpful if the same wording was used for the LCTs throughout the paper.

AR: The figure legend is edited as suggested, thanks for noting it. We considered the close-view map, but the land-cover classification is too coarse in a sense that the chamber locations would not appear properly in a close-view image. That is because the used full-lambda schedule segmentation is region-based and the pixels are merged with the help of spectral (mean pixel value in the segment), textural (SD of pixel values in the segment), shape (areal complexity of the segment) and size information, which we weighted by 0.7, 0.7, 0.3 and 0.3, respectively. The average size of the segment (i.e. pixel: segment ratio) was set to 50 (i.e. 200 m2). The EC footprint (cumulative 90%) is about the area shown in the map and we added a reference to Tuovinen et al. (2019) where the climatology is presented.

Figure 2:

Maybe the use of different symbols for the years would be easier to distinguish for color-blinds. In figure 2f the different lines are hard to tell apart, especially where they are overlapping. Which line is for dry fen, which one for meadow?

AR: Grey-scale colors and different line types were applied. Meadow and dry fen had similar thaw depths; however, the lines were edited for clarity.

Figure 5: Differences between the different months are shown in the figure but not discussed in the text and they do not contribute to the study results. The temporal aspect is interesting but maybe beyond the scope of the study. Figure 6 would be sufficient to answer the research question. Furthermore, the data from different months do not really show an annual course of the CH4 exchange since the data was collected in different years with different meteorological conditions.

AR: That's true. We removed the monthly data.

Figure 6c): It would be helpful if the markers had different colors for the different LCTs.

AR: The figure was edited; temporal data are removed and the spatial data are appended. LCTs are marked with symbols.

Figure 7a): Why is there a vertical line around 50% for the northern wind sector?

AR. Thanks for noticing. There's something odd compared to the original figure, likely produced by the pdf conversion.

---

## Author Response (AR2)

Dear editor,

we want to thank you and both reviewers for the time and helpful comments. We were not able to conduct language revision by a professional native speaker in short notice. We carefully checked the text throughout by the author team. In addition, we understood that BG applies English language copy-editing before producing the galley proofs. Please, let us know if the language edition is still needed. Below, see our point-by-point responses to the reviewers' comments. We are submitting hereby our revised manuscript (track-changes and a changes-accepted copies). Please, note that the line numbers are not matching in the two versions because of the extra lines in the track-changes copy.

Sincerely,

Sari Juutinen on behalf of all authors

**Anonymous referee #2**

The authors have incorporated most suggestions of the first round of reviews into their manuscript as far as the study design / data set allows.

Especially the revised parts of the manuscript now require a language revision since they contain quite some grammar and spelling mistakes.

For example:

l. 40 remove "a" in front of "NEE800 and Pg800"

l. 42 remove one "and"

l. 43 "…the dominant source…"

l. 190 "standard gas" ?

l. 354 "…equally high emissions as the fens…"

l. 503 "nevertheless"

I would suggest a proof-reading of the manuscript by an English native speaker.

A: We did a language editing of the text incl. the text in the above-mentioned cases. There are many small editorial changes (see the track-change copy).

**Anonymous referee #1**

The authors have carefully addressed most of my comments and provide a much improved manuscript version. I do have a few minor comments remaining that I would ask the authors to consider in their final version.

1) the reviewer response states that CH4 flux in barren and all the consuming LCTs was revised and were 6% - 8% of emissions. The modified manuscript text reports 9%. Which one is correct?

2) I remain skeptical if reporting the contribution of CH4 consumption vs. emissions is meaningful considering it is based on such few spatially distributed points (1-2 according to table 1 except for fens and bogs). The authors acknowledge this which I appreciate, but I would encourage them to briefly mention also the following points in the discussion or conclusion:

- seasonal bias of measurements: most measurements are from peak summer and the later half of the growing season (higher temperatures, deeper thaw, active vegetation). I would expect this to result in higher CH4 emissions (also due to plant CH4 transport) but also higher rates of CH4 consumption compared to spring and early summer (lower temperature, shallow thaw, higher soil moisture). Therefore, it would seem to me the reported estimate is the maximum summer contribution of consumption vs. emissions.

- importance of temperature on all gases could be briefly mentioned, even though it is not included in the DCA (not consistently measured).

A: There's variation in the percentage value due to what area was in the focus (barren only, lichen tundra, or all LCTs that consumed $CH_4$). As suggested, we edited the text in the abstract, results and discussion to specify that the estimates represent only growing season. Our chamber data showed large consumption of atmospheric $CH_4$ in tundra barrens and that is supported by the EC data and analysis by Tuovinen et al. (2019). The high rate can be a local feature and related to soil and parent material characteristics.

3) Throughout: please provide number of n when reporting standard error. Otherwise report standard deviation. Since fluxes, especially of CH4 display a high variability, I recommend adding the median, as well as upper and lower quartiles. Reporting only means may overestimate fluxes in this case.

A: We replaced the Table 3 with more through table giving LCT means, medians and standard deviations based on collar specific estimates of $Pg_{800}$, ER, $NEE_{800}$ and collar-specific temporal means of CH4. Those were used to calculate the spatially weighted average $CO_2$ and $CH_4$ fluxes for the landscape (35.8 km2) and proportions of each LCT in it (%). The current table consists also LCT specific means, medians, and standard deviations for $CH_4$ measured during all study years. We ended to the extended table to avoid massive amounts of data figures in the text. The text is edited accordingly. Due to the editions in the Table 3, we edited the table 2 by removing number of observation points (Now in the Table 3). Table 3 is referred in the figure legends when appropriate.